# On The Structure of Parametric Tournaments with Application to Ranking from Pairwise Comparisons

**Vishnu Veerathu**[*]
Cohesity Inc.
ofvishnuveerathu@gmail.com

**Arun Rajkumar**
Indian Institute of Technology, Madras
Robert Bosch Center for Data Science and AI
arunr@cse.iitm.ac.in

## Abstract

We consider the classical problem of finding the minimum feedback arc set on tournaments (MFAST). The problem is NP-hard in general and we study it for important classes of tournaments that arise naturally in the problem of learning to rank from pairwise comparisons. Specifically, we consider tournaments classes that arise out of parametric preference matrices that can lead to cyclic preference relations. We investigate their structural properties via forbidden sub tournament configurations. Towards this, we introduce *Tournament Dimension* - a combinatorial parameter that characterizes the size of a forbidden configuration for rank $r$ tournament classes i.e., classes that arise out of pairwise preference matrices which lead to rank $r$ skew-symmetric matrices under a suitable link function. Our main result is a polynomial-time algorithm - `Rank2Rank` - that solves the MFAST problem for the rank 2 tournament class. This is achieved via a geometric characterization that relies on our explicit construction of a forbidden configuration for this class.

Building on our understanding of the rank-2 tournament class, we propose a very general and flexible parametric pairwise preference model called the local-global model which subsumes the popular Bradley-Terry-Luce/Thurstone classes to capture locally cyclic as well as globally acyclic preference relations. We develop a polynomial-time algorithm - `BlockRank2Rank`- to solve the MFAST problem on the associated Block-Rank 2 tournament class.

As an application, we study the problem of learning to rank from pairwise comparisons under the proposed local-global preference model. Exploiting our structural characterization, we propose `PairwiseBlockRank` - a pairwise ranking algorithm for this class. We show sample complexity bounds of `PairwiseBlockRank` to learn a good ranking under the proposed model. Finally, we conduct experiments on synthetic and real-world datasets to show the efficacy of the proposed algorithm.

## 1 Introduction

A tournament is a complete directed graph. Given a tournament $\mathbf{T}$, the classical feedback arc set on tournament (MFAST) problem asks for the minimum number of edges that must be removed (or whose orientation reversed) in $\mathbf{T}$ to make it acyclic [3]. The problem is known to be NP-hard for general tournaments [6]. We investigate the MFAST problem for several classes of tournaments which naturally occur in learning to rank from pairwise comparisons. In particular, we wish to study the MFAST problem on tournament classes which arise out of parametric pairwise preference classes.

Popular parametric preference models such as the Bradley-Terry-Luce (BTL) [4; 17] and Thurstone models [25] give rise to acyclic tournament matrices for which the MFAST problem is trivial. Our

---

[*]Cohesity, Inc, has neither endorsed nor sponsored this research or its publication in any way; the author's research opinions and findings are his own.

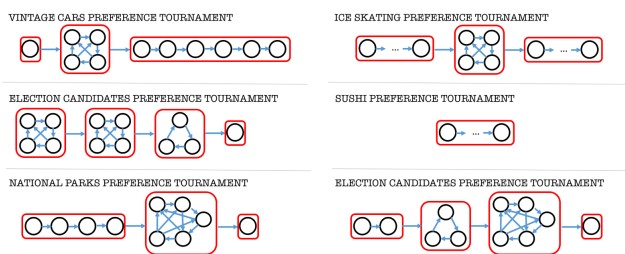

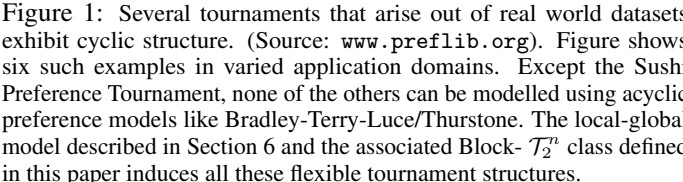

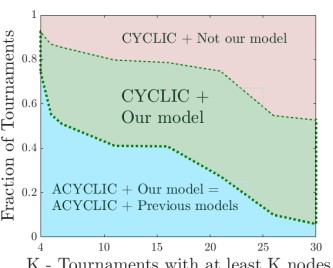

Figure 1: Several tournaments that arise out of real world datasets exhibit cyclic structure. (Source: www.preflib.org). Figure shows six such examples in varied application domains. Except the Sushi Preference Tournament, none of the others can be modelled using acyclic preference models like Bradley-Terry-Luce/Thurstone. The local-global model described in Section 6 and the associated Block-$\mathcal{T}_2^n$ class defined in this paper induces all these flexible tournament structures.

Figure 2: Out of 290 real world tournaments from www.preflib.org, figure shows the fraction of tournaments that are acyclic and the fraction of cyclic tournaments that satisfy the Block-Rank 2 model we propose.

goal is to identify and study non-trivial classes of parametric preference matrices which give rise to potentially cyclic tournaments. Cyclic tournaments arise naturally in several real world preference datasets. Figure 1 shows a few real world cyclic tournament structures and Figure 2 shows the fraction of tournaments out of 290 real world tournaments that are acyclic and those that are cyclic and can arise out of the models proposed in this paper. Clearly, simple models such as BTL/Thurstone are insufficient to model these tournaments.

Every pairwise preference matrix $\mathbf{P} \in (0,1)^{n \times n}$ can be associated to a tournament via a skew symmetric link function $\phi$ i.e, $\phi(\mathbf{P})$ is a skew symmetric matrix and the tournament $\mathbf{T}$ associated to $\mathbf{P}$ has an edge from $i$ to $j$ if and only if $\phi(P_{ij}) > 0$. While preference matrices themselves are always of high rank [11], several well studied pairwise preference classes give rise to skew symmetric matrices of low rank under a suitable link function. For example, the BTL model gives rise to a rank 2 skew symmetric matrix under the `logit` link whereas the Thurstone Model does so under a `probit` link [22]. It is important to understand the possible tournaments any parametric preference class can give rise to. Indeed, it might be natural and much easier for a modeller/domain expert to make a structural statement such as *The class of tournaments of interest should contain cycles of length at most* $4$ as opposed to an algebraic statement like *The class of tournaments should be associated with a preference matrix which on a skew symmetric transformation leads to a rank* $8$ *matrix*.

We study the structure of rank $r$ tournament class i.e, tournaments which arise out of preference matrices that lead to some rank $r$ matrix under a skew-symmetric transformation. This class can be interpreted as each node in the tournament having an embedding in $\mathbb{R}^r$ and the preference relation between nodes depends on a suitable notion of similarity between the embeddings (see Section 4 for details). It is known that such classes can model cyclic relations [22]. To investigate the MFAST problem for the rank $r$ tournament classes, we first structurally understand these tournament via the notion of forbidden configurations [8]. Forbidden configurations in tournaments have been studied in other non-parametric contexts. The simplest example is the class of acyclic tournaments which has the 3 cycle as a forbidden configuration. A complicated forbidden configuration class using Möbius ladders was studied in [8; 9]. In both these cases, the MFAST problem is poly-time solvable.

**Our Contributions.** Our first contribution is to derive upper and lower bounds for the size of a forbidden configuration for any rank $r$ tournament class. We do this by introducing a novel combinatorial parameter that we call the *Tournament Dimension*. When rank $r = 2$, we completely characterize the associated tournament class by identifying the exact forbidden configuration. We use this forbidden configuration to understand the geometry of this tournament class and propose an algorithm (`Rank2Rank`) to solve the MFAST problem in polynomial time for this class of tournaments.

Our next contribution is to propose a flexible class of tournaments called the Block Rank 2 class which builds on the rank 2 class. Tournaments in this class can arise from the *local-global* model - a general pairwise preference model that we propose. The model can capture locally cyclic relations as well as globally acyclic relations and is well suited for several practical applications including those in Figure 1. The model subsumes several popular pairwise ranking models including the BTL and

Thurstone models. We show that the MFAST problem on the associated Block Rank 2 tournament class is polynomial time solvable and propose an algorithm (`BlockRank2Rank`) for the problem.

Our third contribution is to study an application of our work to the problem of learning to rank from pairwise comparisons. We propose a matrix completion based algorithm called `PairwiseBlockRank` for this problem and derive sample complexity bounds for the same under the proposed local-global preference model. Finally, we evaluate our algorithms on real world and synthetic data-sets and show improved performance over several existing algorithms.

## 2 Related Work

Several works have considered learning from transitive pairwise models especially focusing on the Bradley-Terry-Luce (BTL) model[12; 20; 21; 24; 14]. As we focus on cyclic relations, we discuss only the models which can lead to cyclic preferences. While the BTL model can be seen as using a 1 dimensional embedding of the item using a score vector, studies have considered higher dimensional embeddings. A 2 dimensional embedding was considered in [5]. While their model can give rise to cyclic tournaments, it is not clear what type of cycles are possible. [18; 19] propose the Majority vote model which is a random utility model (RUM) with a $d$ dimensional feature embedding for each item. The Majority vote model is powerful enough to produce arbitrarily long cycles and can express any probability sub-matrix over a fixed triplet. The Rank 2 model we study is as powerful as the Majority vote model. Furthermore, we give a complete structural characterization of the tournaments of our model which is not known for the Majority vote model. The Blade-Chest inner (BCI) model [7] embeds each item into two $d$ dimensional vectors (blade vector and a chest vector) and a score vector $\mathbf{s}$ where the probability of $i$ being preferred over $j$ depends on $< i_{\text{chest}}, j_{\text{blade}} > - < i_{\text{blade}}, j_{\text{chest}} > + s_i - s_j$. The rank-2 model can be seen as a special case of the BCI model where $d = 1$ and $\mathbf{s}$ is a constant vector. When $d$ is $\mathcal{O}(n)$, the BCI model can give rise to any tournament as there are $\mathcal{O}(n^2)$ parameters and so the MFAST problem becomes intractable. We focus on the special case of $d = 1$ where there are only $2n$ parameters and propose algorithms to obtain optimal rankings. Finally, we discuss the low rank pairwise rank (LRPR) class of models which result in a low rank matrix under a transformation using a suitable link function [22]. Previous work [12,22] have proposed matrix completion based algorithms to obtain optimal ranking for the LRPR type models assuming transitivity of preferences. In this work, we make no such assumptions.

## 3 Preliminaries

A Tournament is a complete directed graph. We use $i \succ_{\mathbf{T}} j$ to denote that there is a directed edge from node $i$ to $j$ in the tournament $\mathbf{T}$. We call a set of edges $F_\sigma$ in a tournament $\mathbf{T}$ as the feedback arc set of permutation $\sigma$ w.r.t $\mathbf{T}$ if $\sigma(i) > \sigma(j)$ ($j$ ranked ahead of $i$ in $\sigma$) but $i \succ_{\mathbf{T}} j$. Indeed, if one reverses the orientation of the $F_\sigma$ edges in $\mathbf{T}$, one gets an acyclic tournament whose topological sort would yield $\sigma$. We call a permutation $\sigma^*$ as *MFAST-Optimal* for $\mathbf{T}$ if $\sigma^* \in \arg \min_\sigma |F_\sigma|$.

We call $\mathbf{P} \in (0,1)^{n \times n}$ a pairwise preference matrix if $P_{ij} + P_{ji} = 1 \ \forall i \neq j$. We assume $P_{ij} \neq 0.5 \ \forall i, j$. We will say $\mathbf{T_P}$ is a tournament associated with $\mathbf{P}$ if a directed edge from node $i$ to $j$ is present in $\mathbf{T_P}$ if and only if $P_{ij} > 0.5$. We call $\mathbf{P}$ a *rank $r$ preference matrix w.r.t a link function* $\phi$ if $\text{rank}(\phi(\mathbf{P})) = r$ where the function $\phi$ is applied element-wise to $\mathbf{P}$. We will call $\phi$ a skew symmetric link function if $\phi(\mathbf{P})$ is a skew symmetric matrix for any pairwise preference matrix $\mathbf{P}$. Examples of such $\phi$ include the `logit` and `probit` links where $\text{logit}(x) = \log(x/(1-x))$ and $\text{probit}(x) = \Phi^{-1}(p)$ where $\Phi$ is the standard normal cdf. We use $\mathcal{T}_r^{n,\phi}$ to denote the class of all tournaments on $n$ nodes which are associated with a rank $r$ preference matrix w.r.t $\phi$.

Our goal is the study the tournament class $\mathcal{T}_r^{n,\phi}$ where $\phi$ is a skew symmetric link function. Skew-symmetric matrices are naturally associated to tournaments where the edge directions in the tournament are determined by the sign of the corresponding matrix entry. Our results would apply for any skew symmetric $\phi$ and so we will drop the $\phi$ in $\mathcal{T}_r^{n,\phi}$ when it is clear from the context. As skew symmetric matrices have even rank, $\mathcal{T}_r^{n,\phi}$ is non-empty only for even $r$. We wish to investigate the structural constraints on these tournament classes that arise out of the algebraic (rank) restriction that define them. We note that it was known earlier that for the special case when $r = 2$, $\mathcal{T}_2^{n,\phi}$ contains both cyclic and acyclic tournaments [22]. However, nothing further was known about this class.

**Algorithm 1** `Tournament Game`

---

1: **Input:** Integers $k, d$
2: Player-1 chooses a labelled dataset $\{(\mathbf{x}_1, y_1), \ldots (\mathbf{x}_k, y_k)\}$ where $\mathcal{D} = \{\mathbf{x}_i\}_{i=1}^k \in \mathbb{R}^d$ induces a Tournament, $y_i \in \{+1, -1\}$
3: Player-2 chooses a Tournament preserving mapping $f$ w.r.t $\mathcal{D}$
4: **if** $\exists \mathbf{w} \in \mathbb{R}^d$ such that $\text{sign}(\mathbf{w}^T f(\mathbf{x}_i)) = y_i \ \forall i \in [k]$ **then**
5:      Player-2 wins the game
6: **else**
7:      Player-1 wins the game
8: **end if**

---

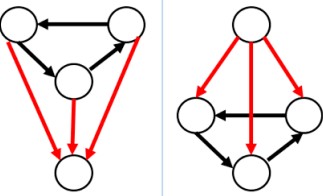

Figure 3: Forbidden configurations for any induced tournament on 4 nodes in $\mathcal{T}_2^n$

**Definition 1.** *A class of tournaments $\mathcal{T}$ is said to have a forbidden configuration of size $\ell$ if $\ell$ is the smallest integer such that there exists a tournament $\mathbf{T}^\ell$ on $\ell$ nodes which does not appear as an induced sub-tournament for any $\mathbf{T} \in \mathcal{T}$*

In other words, no induced sub-tournament on $l$ nodes of $\mathbf{T} \in \mathcal{T}$ is isomorphic to $\mathbf{T}^\ell$. For example, the class of all acyclic tournaments $\mathcal{T}^{\texttt{acyclic}}$ has $C^3$ - the 3-cycle as a forbidden configuration. Taking advantage of this forbidden configuration, one can immediately come up with an algorithm to solve the MFAST problem on $\mathcal{T}^{\texttt{acyclic}}$ - indeed, a topological sort solves this problem. Our interest is to understand if there are similar forbidden configurations for the class $\mathcal{T}_r^{n,\phi}$. Towards this, we start the next section by defining a novel combinatorial parameter called the Tournament dimension.

## 4 Tournament Dimension

Let $d$ be an even integer. We say a set of points $\mathcal{D} = \{\mathbf{x}_1, \ldots, \mathbf{x}_k\} \in \mathbb{R}^d$ *induces a tournament* $\mathbf{T}$ if $\mathbf{x}_i$ is a $d$-dimensional embedding of node $i$ and a directed edge from node $i$ to $j$ exists in $\mathbf{T}$ if and only if $\mathbf{x}_i^T A^{\texttt{rot}} \mathbf{x}_j > 0$. Here, $A^{\texttt{rot}} \in \mathbb{R}^{d \times d}$ is the block diagonal matrix with $\frac{d}{2}$ blocks where each block of size 2 is a rotation matrix $[0 \ -1; 1 \ 0]$ [2]. Note that $\mathbf{x}_i^T A^{\texttt{rot}} \mathbf{x}_j = -\mathbf{x}_j^T A^{\texttt{rot}} \mathbf{x}_i \ \forall i, j$ and the tournament inducing property of $\mathcal{D}$ excludes datapoints such that $\mathbf{x}_i^T A^{\texttt{rot}} \mathbf{x}_j = 0$. We note that $d$ being even is not a restriction on the embedding. If $d$ is odd, one can increase the embedding to $d + 1$ dimensions w.l.o.g where the last component is 1 for all $\mathbf{x}_i$.

**Lemma 1.** *Let $\mathcal{D} = \{\mathbf{x}_1, \ldots, \mathbf{x}_k\} \in \mathbb{R}^d$ induce a tournament $\mathbf{T}$. Then $\mathbf{T} \in \mathcal{T}_d^k$. Furthermore, for every $\mathbf{T} \in \mathcal{T}_d^k$ there exists a dataset $\mathcal{D}$ with $k$ vectors in $\mathbb{R}^d$ that induces $\mathbf{T}$.*

We call a mapping $f : \mathbb{R}^d \to \mathbb{R}^d$ as *Tournament-preserving* w.r.t. a tournament inducing dataset $\mathcal{D}$ if $(f(\mathbf{x}_i)^T A^{\texttt{rot}} f(\mathbf{x}_j) > 0) \iff (\mathbf{x}_i^T A^{\texttt{rot}} \mathbf{x}_j > 0) \ \forall \mathbf{x}_i, \mathbf{x}_j \in \mathcal{D}$. We define the *Tournament game* between two players as described in Algorithm 1.

**Definition 2.** *The Tournament dimension for $d$ - `TourDim`$(d)$ - is the largest value of $k$ for which Player 2 always has a winning strategy in the Tournament game (Algorithm 1).*

**Remark.** `TourDim`$(d) = k$ implies that Player 1 has a winning strategy for all values $\geq k + 1$. In particular, there is a tournament $\mathbf{T} \in \mathcal{T}_d^k$ induced by the dataset $\mathcal{D}$ and a labelling $\{y_1, \ldots, y_k\}$ strategically chosen by Player 1 such that there does not exist a $\mathbf{T}$-preserving mapping for which Player 2 can find a $\mathbf{w} \in \mathbb{R}^d$ that labels all points in $\mathcal{D}$ correctly. The inability to find such a $\mathbf{w}$ is equivalent to the inability to add one more node to $\mathbf{T}$ whose edge direction to node $i$ of $\mathbf{T}$ is specified by the label $y_i$. Thus, there must be a forbidden configuration on $k + 2$ nodes.

The Theorem below formalizes the above remark.

**Theorem 2.** *$\mathcal{T}_r^n$ has a forbidden configuration of size `TourDim`$(r) + 2$ for every even integer $r < n$ and every $n \geq$ `TourDim`$(r) + 2$.*

---

[2]The matrix $A^{\texttt{rot}}$ is fundamental to skew-symmetric matrices. In fact, any skew-symmetric bilinear form can be represented under a suitable basis using $A^{\texttt{rot}}$ and additional zero diagonal blocks as necessary [23].

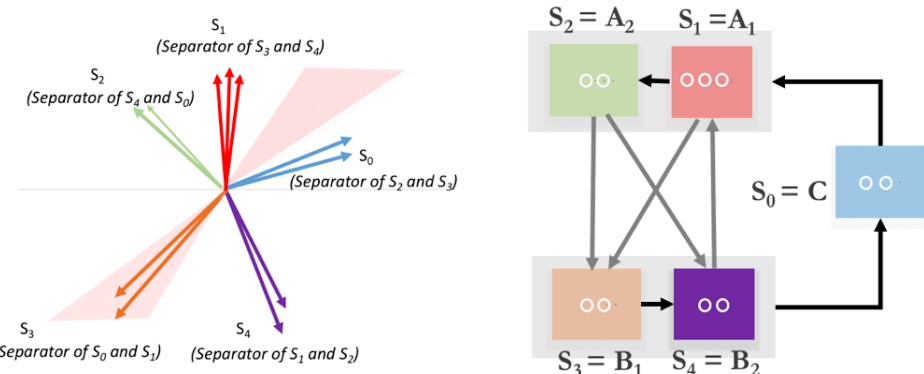

Figure 4: (Left) Geometry of the embeddings of nodes in $\mathcal{T}_2^n$ where $-S_3 \subset \texttt{cone}(S_0 \cup S_1)$ is highlighted. (Right) Structure of Tournament belonging to $\mathcal{T}_2^n$ for the example on the left. The sets $\{S_0, \ldots, S_4\}$ are colored as per the left figure and mapped to $\{A_1, A_2, B_1, B_2, C\}$ as in Theorem 6.

We bound the size of the forbidden configuration as follows

**Theorem 3.** *Let $\mathcal{T}_r^n$ have a forbidden configuration of size $\ell$. Then $r + 2 \leq \ell \leq 2^r + r + 1$*

**Remark.** *We believe that the upper bound in Theorem 3 is not tight in general and leave it as an open problem to improve this for a general $r$.*

**Corollary 1.** *$\mathcal{T}_2^n$ has a forbidden configuration of size $4$.*

While the above Corollary gives the size of the forbidden sub configuration, it does not specify what exactly the configuration is. As the rank 2 class occurs naturally in several learning to rank problems, we investigate this further in the next section.

# 5    Rank $2$ Tournaments - $\mathcal{T}_2^n$

In this section we focus on the Rank 2 tournament class $\mathcal{T}_2^n$. To motivate this class, consider the following generalisation of the popular Bradley-Terry-Luce (BTL) preference model where for a skew symmetric link function $\phi$, we define $\phi(P_{ij}) = u_i v_j - v_i u_j$. The model is parameterized by two vectors $\mathbf{u}, \mathbf{v} \in \mathbb{R}^n$. When one of these vectors is the all ones vectors and when the link function $\phi$ is the $\texttt{logit}$ function, the model reduces to the standard BTL model.

In general for any $\mathbf{P}$, if $\phi$ is a skew symmetric link function that results in a rank 2 matrix, then $\phi(\mathbf{P}) = \mathbf{u}\mathbf{v}^T - \mathbf{v}\mathbf{u}^T$ for some $\mathbf{u}, \mathbf{v} \in \mathbb{R}^n$. Such a model can be interpreted as each node $i$ having a two dimensional embedding $\mathbf{h}_i = [u_i, v_i]$. Here, the direction of the edge between nodes $i$ and $j$ in the associated tournament depends on the sign of $u_i v_j - v_i u_j = \mathbf{h}_i^T A^{\texttt{rot}} \mathbf{h}_j$ where $A^{\texttt{rot}} \in \mathbb{R}^{2 \times 2}$ is the rotation matrix $[0 - 1; 1 \, 0]$. With this background, we now proceed to characterize the class $\mathcal{T}_2^n$.

**Geometric Characterization of $\mathcal{T}_2^n$:** We start with the following key lemma:

**Lemma 4.** *(**Forbidden configurations**) Let $n \geq 4$. Every induced tournament of $\mathcal{T}_2^n$ on a subset of four nodes forbids the configurations in Figure 3.*

The above lemma can be used to geometrically characterize the two dimensional embeddings of the nodes in $\mathcal{T}_2^n$. For this characterization, we need the following definition.

We call a set of vectors $S \subset \mathbb{R}^2$ a *separator* of two sets of vectors $U, V \subset \mathbb{R}^2$ if the convex cone of $U \cup V$ contains $-S$ i.e., the set of vectors in $S$ rotated by 180 degrees is a subset of $\texttt{cone}(U \cup V)$

**Theorem 5.** *Let $\mathbf{T} \in \mathcal{T}_2^n$ where each node $i$ has a two dimensional embedding $\mathbf{h}_i \in \mathbb{R}^2$. Then either $\mathbf{T} \in \mathcal{T}^{\texttt{acyclic}}$ or $\exists k \geq 1$ such that the vectors $\mathbf{h}_i$ can be partitioned into $2k + 1$ ordered sets $\{S_0, \ldots, S_{2k}\}$ where for each $i = 0, \ldots, 2k, S_i \subset \mathbb{R}^2$ is a separator of the sets $S_{i+k \bmod (2k+1)}$ and $S_{i+k+1 \bmod (2k+1)}$*

Figure 4 (left) shows how the embeddings of nodes that result in $\mathcal{T}_2^n$ would look like geometrically. As one can observe, the example tournament has 5 sets of vectors and every set of vectors of the

same color acts as a separator for two other sets of different colors. This is the crucial geometric insight that we obtain which characterizes this tournament class. We note that when the number of sets (colors) is just 1, the model leads to acyclic tournaments. Indeed, tournaments arising out of the BTL/Thurstone models have this property and lead to $\mathbf{T} \in \mathcal{T}^{\mathtt{acyclic}} \subsetneq \mathcal{T}_2^n$. However, $\mathcal{T}_2^n$ is much more richer than the simple $\mathcal{T}^{\mathtt{acyclic}}$ class.

**Structural Characterization of $\mathcal{T}_2^n$:** With the help of the geometric characterization, we next give the structural description of all tournaments in $\mathcal{T}_2^n$. For two sets of nodes $A$ and $B$, We use $A \succ_{\mathbf{T}} B$ to denote that every node in $A$ has an outgoing edge to every node in $B$ in the tournament $\mathbf{T}$. Also, $A \prec_{\mathbf{T}} B$ whenever $B \succ_{\mathbf{T}} A$

**Theorem 6.** *Let $n \geq 4$ and $\mathbf{T} \in \mathcal{T}_2^n$. Then either $\mathbf{T} \in \mathcal{T}^{\mathtt{acyclic}}$ or there exists $k \geq 1$ such that the $n$ nodes can be partitioned into $2k + 1$ sets $\{A_1, A_2, \ldots, A_k, B_1, B_2, \ldots, B_k, C\}$ where*
*(a) The induced sub-tournament on each of the $2k + 1$ sets belong to $\mathcal{T}^{\mathtt{acyclic}}$*
*(b) $A_i \succ_{\mathbf{T}} A_j, B_i \succ_{\mathbf{T}} B_j \quad \forall i < j$*
*(c) $A_i \succ_{\mathbf{T}} B_j \, \forall i \geq j, A_i \prec_{\mathbf{T}} B_j \, \forall i < j$*
*(d) $B_i \succ_{\mathbf{T}} C \succ_{\mathbf{T}} A_j \quad \forall i, j$*

Figure 4 (right) gives a graphical representation of the structure of $\mathcal{T}_2^n$ described in Theorem 6. The following theorem sheds light on the rich expressivity of the $\mathcal{T}_2^n$ tournament class.

**Theorem 7.** *(**Long Cycles and Arbitrary Triplets**) Given any arbitrary ordered set of indices $\{i_1, i_2, \ldots, i_k\}, k \leq n$, there exists a $\mathbf{T} \in \mathcal{T}_2^n$ such that $i_1 \succ_{\mathbf{T}} i_2 \succ_{\mathbf{T}} \ldots \succ_{\mathbf{T}} i_k \succ_{\mathbf{T}} i_1$. Furthermore, Given any $p_1, p_2, p_3 \in (0, 1)$ and indices $i_1, i_2, i_3 \in [n]$, there exists a pairwise preference matrix $\mathbf{P}$ whose associated $\mathbf{T_P} \in \mathcal{T}_2^n$ where $\mathbf{P}(i_1, i_2) = p_1, \mathbf{P}(i_2, i_3) = p_2, \mathbf{P}(i_3, i_1) = p_3$.*

### 5.1 Polynomial Time Algorithm for MFAST for $\mathcal{T}_2^n$

We present a polynomial time algorithm for MFAST for the $\mathcal{T}_2^n$ class. The algorithm Rank2Rank (Algorithm 2) takes as input a tournament $\mathbf{T}$ and outputs a permutation $\sigma^*$ that is MFAST-optimal for $\mathbf{T}$ whenever $\mathbf{T} \in \mathcal{T}_2^n$. The algorithm finds such a $\sigma^*$ by relying crucially on the structural understanding gained in the previous section via forbidden configurations. In the following, given a set of nodes $A$ and a tournament $\mathbf{T}$, we represent the induced tournament on $A$ by $\mathbf{T}$ as $\mathbf{T}_A$.

---

**Algorithm 2** Rank2Rank-R2R

1: Input: A tournament $\mathbf{T}$
2: **if** $\mathbf{T} \in \mathcal{T}^{\mathtt{acyclic}}$ **then**
3:     Obtain $\sigma*$ by a Topological sort of $\mathbf{T}$
4: **else**
5:     Find $\{a, b, c\}$ s.t $(a \succ_{\mathbf{T}} b \succ_{\mathbf{T}} c \succ_{\mathbf{T}} a)$
6:     $A := \{a\} \cup \{i : i \succ_{\mathbf{T}} b \text{ and } c \succ_{\mathbf{T}} i\}$
7:     $B := \{b\} \cup \{i : i \succ_{\mathbf{T}} c \text{ and } a \succ_{\mathbf{T}} i\}$
8:     $C := \{c\} \cup \{i : i \succ_{\mathbf{T}} a \text{ and } b \succ_{\mathbf{T}} i\}$
9:     $\sigma_0 = [\text{R2R}(\mathbf{T}_A), \text{R2R}(\mathbf{T}_B), \text{R2R}(\mathbf{T}_C)]$
10:     Let $\sigma_h$ be the permutation obtained by a cyclic shift of $\sigma_0$ by $h$ positions.
11:     Let $\sigma^*$ be the permutation among $\sigma_h \, \forall h$ which has the least size of the feedback arc set w.r.t $\mathbf{T}$ i.e., $\sigma^* = \arg \min_h |F_{\sigma_h}|$

12: **end if**
13: Output $\sigma^*$

---

**Algorithm 3** BlockRank2Rank - BR2R

1: Input: A tournament $\mathbf{T}$
2: **if** $\mathbf{T} \in \mathcal{T}^{\mathtt{acyclic}}$ **then**
3:     Obtain $\sigma^*$ by a Topological sort of $\mathbf{T}$
4: **else**
5:     Find $\{a, b, c\}$ s.t $(a \succ_{\mathbf{T}} b \succ_{\mathbf{T}} c \succ_{\mathbf{T}} a)$
6:     $S^+ = \{i : i \succ_{\mathbf{T}} \{a, b, c\}\}$
7:     $S^- = \{i : i \prec_{\mathbf{T}} \{a, b, c\}\}$
8:     $S = [n] \backslash \{S^+ \cup S^-\}$
9:     $\sigma^* = [\text{BR2R}(\mathbf{T}_{S^+}), \text{R2R}(\mathbf{T}_S), \text{BR2R}(\mathbf{T}_{S^-})]$
10: **end if**
11: Output $\sigma^*$

---

**Theorem 8.** *If $\mathbf{T} \in \mathcal{T}_2^n$ is given as input to the Rank2Rank (Algorithm 2), the output $\sigma^*$ produced by the algorithm is MFAST-optimal for $\mathbf{T}$. Furthermore, Rank2Rank has a time complexity of poly($n$).*

**Geometric Sweep Interpretation:** One can also come up with a geometric sweep algorithm to solve the MFAST problem for $\mathcal{T}_2^n$ inspired by the geometric embedding based interpretation given earlier. Here, we assume that the embeddings of the items are given as input. The algorithm begins by fixing an arbitrary anchor item $i$ and circular sweeps the $\mathbb{R}^2$ plane in the counter clockwise direction from the embedding of $i$ thus creating a permutation $\sigma_0$ of items corresponding to the order in which they

are encountered in the sweep. Starting to sweep from different anchors would give rise to $n$ different cyclic shifts of $\sigma_0$. We argue in the appendix that the optimal $\sigma^*$ that solves the MFAST problem w.r.t $\mathbf{T}$ must necessarily be one of these $n$ permutations. We note that while the Geometric sweep algorithm needs the embeddings to be known, the `Rank2Rank` algorithm does not need the same.

# 6 Block $\mathcal{T}_2^n$ Class

Often in practical applications, items can be clustered into blocks such that within block they exhibit a certain *local* pairwise preference whereas across blocks they exhibit a *global* pairwise preference. For example, consider the game of tennis where the top 3 players might have a cyclic preference among themselves whereas they strictly beat every other player in the bottom $n - 3$. To capture such effects, we next propose the *local-global* pairwise preference model.

## 6.1 Local-Global Preference Model

The local-global pairwise preference model is parameterized by three vectors $\mathbf{s}, \mathbf{u}, \mathbf{v} \in \mathbb{R}^n$ and a ordered partition of $[n]$ given by $\{H_1, ..., H_b\} \in 2^{[n]}$ where $\cup_{k=1}^b H_k = [n]$ and $H_i \cap H_j = \emptyset \ \forall i \neq j$. Let $g(i)$ denote the partition in which item $i$ is present. We assume that $s_i > s_j \iff g(i) < g(j)$ i.e., the parameter vector $\mathbf{s}$ aligns with the ordering of the partitions. The pairwise preference probability of item $i$ being preferred over item $j$ under a skew symmetric link function $\phi$ is given as

$$\phi(P_{ij}) = \begin{cases} u_i v_j - v_i u_j & \text{if } \exists \ell \text{ s.t } i, j \in H_\ell \\ s_i - s_j & \text{otherwise} \end{cases}$$

Figure 5 shows some flexible tournament structures that arise out of the local global model and Figure 1 shows several real world tournaments that have associated tournaments arising out of this model.

One can interpret the above model as follows: The set of $[n]$ items is divided into $b$ blocks/partitions. Restricted to each block, the *local* pairwise preference matrix has an associated tournament that belongs to $\mathcal{T}_2^n$ which can potentially contain cycles. However, across partitions, the preference matrix behaves like a standard BTL type model parameterized by the score vector $\mathbf{s}$ and hence the associated *global* block level tournament relation belongs to $\mathcal{T}^{\texttt{acyclic}}$. The tournament associated to the entire set of $n$ nodes thus has a *block* structure. Indeed, the model is determined by $3n$ parameters where $2n$ parameters determine the intra block local structure and $n$ parameters the inter block global structure.

**Remark.** *The standard BTL model and it's generalization with $2n$ parameters are special cases of the above model where the number of blocks $b = 1$ and $\phi$ is the `logit` link.*

We refer to the class of tournaments that arise out of a local-global model as the $\texttt{Block} - \mathcal{T}_2^n$ class.

**Theorem 9.** *If $\mathbf{T} \in \texttt{Block-}\mathcal{T}_2^n$ is given as input to the `BlockRank2Rank` (Algorithm 3), the output $\sigma^*$ produced by the algorithm if MFAST-optimal for $\mathbf{T}$. Furthermore, `BlockRank2Rank` has a time complexity of poly($n$).*

# 7 Learning to Rank From Pairwise Comparisons under $\texttt{Block-}\mathcal{T}_2^n$

We now consider the application of the models discussed so far to the problem of learning to rank from pairwise comparisons under the local-global pairwise preference model which lead to tournaments in $\texttt{Block-}\mathcal{T}_2^n$. Towards this, we propose the `PairwiseBlockRank` algorithm (Algorithm 4). Given a dataset of pairwise comparisons, the algorithm constructs the empirical pairwise comparison matrix. It applies the link function $\phi$ to construct a empirical skew-symmetric matrix with missing entries (corresponding to pairs that are not compared in the dataset). The algorithm then applies a matrix completion routine to complete the skew-symmetric matrix. Once complete, the associated tournament is computed and the ranking obtained using the `BlockRank2Rank` algorithm.

Matrix completion routines (such as [16]) typically require an upper bound on the rank of the matrix being completed. The following Lemma establishes that this depends on $b$, the number of blocks.

**Lemma 10.** *Let $\mathbf{T} \in \texttt{Block-}\mathcal{T}_2^n$ with $b$ blocks. Then $\mathbf{T} \in \mathcal{T}_r^n$ for some $r \leq 4b$.*

The following theorem establishes sample complexity of learning for recovering the blocks correctly.

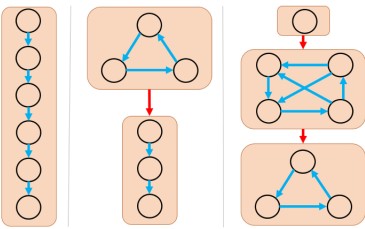

Figure 5: Example of a few Flexible Tournament Structures arising out of the *local-global* model with $1, 2$ and $3$ blocks.

**Algorithm 4** PairwiseBlockRank

1: Input: $\mathcal{D} = \{(i_k, j_k), y_k\}$, $k = \{1, \ldots, m\}$, link function $\phi$, rank $r$
2: Compute Empirical pairwise preference matrix:
3: **if** $(i, j) = (i_k, j_k)$ for some $k$ **then**
4: $\quad \hat{P}_{ij} = \sum_{k=1}^{m} \frac{\mathbb{I}(i_k=i, j_k=j, y_k=1)}{\mathbb{I}(i_k=i, j_k=j)}; \hat{P}_{ji} = 1 - P_{ij}$
5: **else**
$\quad \hat{P}_{ij} = \hat{P}_{ji} = 0$
6: **end if**
7: $\hat{\mathbf{M}} = \texttt{Matrix-Completion}(\phi(\hat{\mathbf{P}}), r)$
8: Construct Tournament $\mathbf{T}$ from $\hat{\mathbf{M}}$ where $(i, j) \in E$ whenever $\hat{M}_{ij} > 0$
9: $\sigma^* = \texttt{BlockRank2Rank - BR2R}(\mathbf{T})$
10: Output $\sigma^*$

**Theorem 11.** (***Block Recovery***) *Let the local global model be parameterized by vectors* $\mathbf{u}, \mathbf{v}, \mathbf{s} \in \mathbb{R}^n$ *and ordered partitions* $\{H_1, \ldots, H_b\}$ *and a skew symmetric link function* $\phi$. *Let* $m$ *pairs be chosen uniformly at random from all subsets of* $\binom{n}{2}$ *of size* $m$ *and let each pair be compared* $K$ *times according to the corresponding preference matrix* $\mathbf{P}$. *Let* $\theta = \min_{ij} |u_i v_j - v_i u_j|$ *where* $(i, j)$ *are in the same partition and let* $\Delta = \min |s_i - s_j|$ *where* $(i, j)$ *are in different partitions. Let* $0 < \epsilon < n^2(\frac{\theta}{\Delta})^2$. *Then with probability at least* $1 - \frac{2}{n^3}$, *if* $m$ *is* $O(nb\log(n)))$ *and* $K$ *is* $O(\frac{b\log(n)}{\epsilon\Delta^2})$, *the* PairwiseBlockRank *algorithm on running with the dataset generated as above, returns a ranking* $\sigma$ *that respects the block structure i.e.,* $\sigma(i) < \sigma(j)$ *whenever* $i \in H_k$ *and* $j \in H_\ell \forall \ell > k$.

# 8 Experiments

**Datasets:** We use the following real world datasets (with $n$ items to be ranked and $m$ pairwise comparisons): DoTA [1] ($n = 757, m = 10442$), Tennis [2] ($n = 742, m = 23806$), Sushi-A [15] ($n = 10, m = 100000$), Sushi-B [15] ($n = 100, m = 25000$), Jester [13] ($n = 100, m = 891404$).

**Setting + Performance measure:** Given a set of pairwise comparisons, we do a $70 : 30$ train:test split. We run the algorithms on the train data to obtain a ranking $\sigma$. We test the performance of $\sigma$ on the test set by computing the ratio of *upsets*(the pairs $(i, j)$ for which $\sigma(j) < \sigma(i)$ but the fraction of times $i$ being preferred over $j$ in the test set is $\geq 0.5$) and the number of unique pairs in the test data.

**Algorithms:**
**MC + Copeland**: We first complete the empirical preference matrix $\hat{\mathbf{P}}$ using a matrix-completion (MC) routine with rank = 2 and then run the standard Copeland procedure [10] to get a ranking.

**MC + Borda**: Same as Copeland but the standard Borda algorithm [21] is used instead of Copeland to obtain a ranking from the completed ranking.

**Rank Centrality**: A spectral ranking algorithm [20] that ranks based on computing a stationary distribution of a Markov chain associated with $\hat{\mathbf{P}}$.

**Blade-Chest**: A maximum likelihood based algorithm for the Blade-Chest model [7] where we set the dimension of the emdedding to be $8$ (Other choices perform poorly, see Supplementary). To obtain a ranking from the estimated MLE, we run the BlockRank2 algorithm of this paper (Other choices perform poorly, see Supplementary).

**PariwiseBlockRank**: The algorithm described in this paper. To make it more suitable for real-world data, in each recursive call, we use either the ranking given by the algorithm or the Copeland procedure depending on whichever is better for the given data.

The results are presented in Table 1 averaged over 20 splits. As can be seen, the PairwiseBlockRank algorithm performs better or comparably with the best algorithm for most of the datasets. In some cases, it produces similar results as the Copeland algorithm. Furthermore, the Blade-Chest model

| Algorithml Dataset | MC + Copeland | MC + Borda | Rank Centrality | Blade-Chest | Pairwise Block-Rank |
|---|---|---|---|---|---|
| DOTA | 0.311 ±0.01 | 0.368 ±0.03 | **0.215** ±0.01 | 0.228 ±0.01 | 0.23 ±0.03 |
| Tennis | 0.314 ±0.005 | 0.314±0.006 | 0.285 ±0.008 | 0.311 ±0.006 | **0.281** ±0.006 |
| Sushi-A | 0.036±0.02 | 0.18 ±0.02 | 0.042 ±0.01 | 0.06±0.01 | **0.034**±0.02 |
| Sushi-B | **0.176**±0.006 | 0.188±0.005 | **0.176** ±0.006 | 0.203±0.006 | **0.176**±0.006 |
| Jester | **0.05** ±0.001 | 0.23 ±0.002 | **0.05** ±0.001 | 0.15 ±0.022 | **0.05** ±0.001 |

Table 1: Results on real-world datasets. Algorithms with the best results are in boldface.

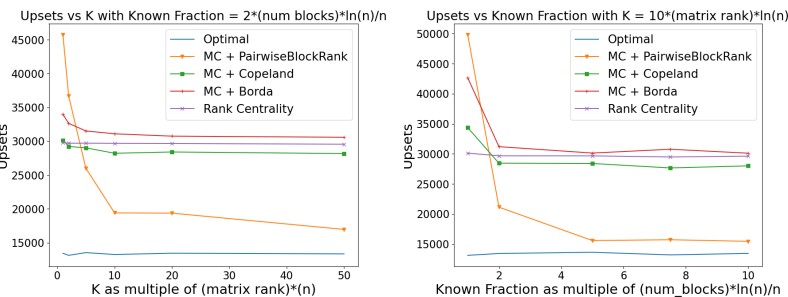

Figure 6: Results for **P** under local-global model with 3 blocks. Refer Section 8 for details.

performs poorly for all datasets. The Rank-centrality algorithm performs competitively. To understand this better, we run the following synthetic experiments to compare with rank centrality algorithm.

**Synthetic Experiments:** The synthetic data was generated to follow the local-global Model. We considered the number of items $n = 600$, and ran experiments on $b = 3$ (see Supplementary for other choices) block cases with equal sized blocks. From the constructed underlying probability preference matrix, we sampled entries according to two parameters, $K$ and known fraction $\beta$. Known fraction ($\beta$) represents the fraction of total unique pairs that are uniformly chosen for comparison and $K$ represents the number of Bernoulli comparisons for every such pair chosen. We did two types of experiments, one where we fixed the known fraction $\beta$ and varied $K$ and other where we fixed $K$ and varied $\beta$. Thus for every experiment $K.\beta.\binom{600}{2}$ comparisons are made in total. Figure 6 shows the results from varying $K$ and the known fraction. The results are averaged across 10 runs. As can be seen, for the MC + `PairwiseBlockRank` algorithm, the pairwise disagreements quickly approach the optimal value when the known fraction is fixed, and $K$ increases. For the other algorithms, the quality of ranking does not improve with increase in $K$. Similar trend is observed when the known fraction $\beta$ is increased while fixing $K$. The results clearly indicate that with increasing samples, the proposed algorithm quickly converges to the optimal ranking whereas the others perform poorly.

## 9 Conclusion

In this work, we study the class of parametric preference matrices which induce cyclic tournaments. We propose algorithms that solve the MFAST problem on rank 2 classes and build on the rank 2 class to propose a flexible model called the local-global model. For the general rank $r$ class, we initiate understanding the associated tournament class via the notion of tournament dimension. Going forward, we would like to understand the forbidden configurations of higher rank tournaments and pin down the difficulty in solving the MFAST problem using this approach.

**Broader Impact** We introduce a flexible parametric model that can capture potentially acyclic relations. The broader impact is that we look at the classical MFAST problem from a novel parametric viewpoint. We believe this can lead to broader impact in fundamental understanding of the hardness of this problem. Algorithms might be subject to bias if the data is inherently biased. The current approach does not focus on removing algorithmic bias that can arise in this fashion.

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
