# A Appendix

**Link to Source Code** An anonymous link to the source code is present here.

**Theorem 12.** *(Transitivity) Let* $\mathbf{P} \in$. *Let* $i, j, k$ *be three items with embeddings* $h_i, h_j, h_k \in \mathbb{R}^2$ *respectively. If there exits a* $d \in \mathbb{R}^2$ *such that* $h_a^T d > 0 \ \forall a \in \{i, j, k\}$, *then* $\mathbf{T_P}(\{i, j, k\})$ *is acyclic.*

*Proof.* Let $\theta_1$ be the counterclockwise angle between $i$ and $j$ and $\theta_2$ be the angle between $j$ and $k$. W.l.o.g assume that $i \succ_{\mathbf{P}} j$ and $j \succ_{\mathbf{P}} k$. Then $0 < \theta_1, \theta_2 < \pi$. The relationship between $i$ and $k$ is determined by $\theta_1 + \theta_2$. However, the existence of a $d \in \mathbb{R}^2$ satisfying the condition of the theorem implies that the embeddings of the three items $\{i, j, k\}$ lie in the same half space. This implies that $0 < \theta_1 + \theta_2 < \pi$ which further implies that $i \succ_{\mathbf{P}} k$. The result follows.

$\square$

**Proof of Lemma 1**

*Proof.* Let $\mathcal{D} = \{\mathbf{x}_1, \ldots, \mathbf{x}_k\}$ induce a tournament $\mathbf{T}$. As $T_{ij} = \text{sign}(\mathbf{x}_i^T A^{\text{rot}} \mathbf{x}_j)$ by definition, $\mathbf{T}$ is associated with the skew symmetrix matrix

**Proof of Theorem 2**

*Proof.* Assume that the $\text{TourDim}(d) = k$. Thus Player 1 has a winning strategy for $k + 1$ i.e., there exists a dataset $\mathcal{D}$ of size $k + 1$ that induces a tournament $\mathbf{T} \in \mathcal{T}_d^{k+1}$ such that Player 2 cannot produce a $\mathbf{T}$ preserving mapping $f$ such that one can find a $\mathbf{w}$ where $f(\mathbf{x}_i) = y_i$ for all $i$. Note that the ability of Player 2 to find such a $\mathbf{w}$ is equivalent to the ability of adding one more node to the tournament $\mathbf{T}$ whose direction with respect to the node $i$ of $\mathbf{T}$ is given by the label $y_i$ where the new tournament would belong to $\mathcal{T}_d^{k+2}$. This is true as this new tournament with the extra node is induced by the dataset $\{\mathcal{D} \cup (A^{\text{rot}})^{-1} \mathbf{w}\}$ (Note that $A^{\text{rot}}$ is invertible). As Player 2 is unable to produce such a mapping and taking into account that both $\mathcal{D}$ (and so $\mathbf{T}$) and $y_i$'s were chosen strategically by Player 1, this implies that no tournament in $\mathcal{T}_d^{k+2}$ can contain $\mathbf{T}$ as an induced sub configuration. Finally consider any $n \geq k + 1$ and let $\mathbf{T} \in \mathcal{T}_d^n$. As $\mathbf{T}$ restricted to any subset of $k + 1$ nodes must belong to $\mathcal{T}_d^{k+1}$, the result follows. $\square$

Theorem 3: The bounds for $\ell$ must be as follows: $r + 2 \leq \ell \leq 2^r + r + 1$ and not $r$ and $2^r$ as stated in the submission.

**Proof of Theorem 3**

*Proof.* By definition $\ell = \text{TourDim}(r) + 2$. Thus we need to show that $\text{TourDim}(r) \geq r$. Assume not. Then Player 1 has a winning strategy for $r$ i.e., there exists a dataset $\mathcal{D}$ of size $r$ such that there does not exist a $\mathbf{T}$ preserving mapping such that one can find a $\mathbf{w}$ with $f(\mathbf{x}_i) = y_i$ for all $i$. However, it is always possible to perturb the datapoints randomly by a small $\epsilon$ amount such that the perturbed $\mathcal{D}$ still induces $\mathbf{T}$ and the data points are linearly independent with probability 1. For such a perturbed dataset, there must exist a $\mathbf{w}$ that achieves the labeling given by $y_i$ - just solve for $\mathbf{Xw} = \mathbf{y}$ where $\mathbf{X} \in \mathbb{R}^{d \times d}$ contains the data points in the perturbed $\mathcal{D}$ as its columns. Thus, we arrive at a contradiction. This proves the lower bound.

To prove the upper bound, fix an arbitrary tournament $\mathbf{T} \in \mathcal{T}_r^{r+1}$ on $r + 1$ nodes where each node has an embedding $\mathbf{x}_i \in \mathbb{R}^r$. Then $\mathbf{x}_{r+1} = \sum_{i=1}^{r} c_i \mathbf{x}_i$ for some constants $c_i$s. Assume wlog that $c_i \neq 0 \ \forall i$ (If some set of $c_i = 0$, then the bound we get would be tighter). Now consider adding an extra node to $\mathbf{T}$ whose directions with the $r + 1$ nodes are given by the label vector $y \in \{\pm 1\}^{r+1}$ where $y_{r+1} = -1$. If this new tournament is not a forbidden configuration for $\mathcal{T}_r^{r+2}$, then there must exist some $z \in \mathbb{R}^r$ such that $\text{sign}(z^T A^{\text{rot}} \mathbf{x}_i) = y_i$ for all $i = 1, \ldots, r + 1$. We argue that such a $z$ invalidates a particular sign pattern for the coefficients $c_i$'s. Indeed, the following is true:

$$z^T A^{\text{rot}} \mathbf{x}_{r+1} = \sum_{i=1}^{r} c_i (z^T A^{\text{rot}} \mathbf{x}_i)$$

$y_{r+1} = \text{sign}(z^T A^{\text{rot}} \mathbf{x}_{r+1}) = -1 \implies \text{sign}(c_i) = y_i \ \forall i$ is not possible as it would lead to a contradiction since $\text{sign}(z^T A^{\text{rot}} \mathbf{x}_i) = y_i$.

As the choice for $y_i$ for $i = 1, \ldots, r$ is arbitrary, one can keep attempting to add nodes to the tournament with all possible sign patterns with respect to the first $r$ nodes. Indeed after adding $2^r$ such nodes exhausting all sign patterns for $y_1, \ldots, y_r$, one invalidates all possible sign patterns for the coefficients $c_1, \ldots c_r$. But this is a contradiction. Thus, starting from an arbitrary tournament on $\mathbf{T} \in \mathcal{T}_r^{r+1}$ we arrive at a forbidden configuration for $\mathcal{T}_r^{r+1+2^r}$ nodes.

Finally, for a general $n \geq r + 1 + 2^r$, if $\mathbf{T} \in \mathcal{T}_r^n$, any induced tournament on $r + 1 + 2^r$ nodes must belong to $\mathcal{T}_r^{r+1+2^r}$ and so the above argument suffices. $\qquad\square$

**Proof of Corollary 1**

*Proof.* From Theorem 3, it follows that the forbidden configuration for $\mathcal{T}_2^n$ must be of size at least 4. We will show that this is indeed exactly equal to 4 by showing that $\texttt{TourDim}(2) = 2$. To show this, we need to demonstrate a strategy for player 1 to win with 3 data points in $\mathbb{R}^2$. Let the data points selected by player 1 induce the 3 cycle as the tournament and the corresponding labels be $y_1 = y_2 = y_3 = 1$. Any $\mathbf{T}$ preserving mapping should necessarily map the datapoints to some $\mathbf{x}_1, \mathbf{x}_2, \mathbf{x}_3 \in \mathbb{R}^2$ where the counterclockwise angle between $\mathbf{x}_1$ and $\mathbf{x}_2$ is less than 180 degrees and $-\mathbf{x}_3 \in \text{cone}\{\mathbf{x}_1, \mathbf{x}_2\}$. If not, one cannot induce the 3-cycle as a tournament. Then for any $\mathbf{w} \in \mathbb{R}^2$ such that $\text{sign}(\mathbf{w}^T\mathbf{x}_1) = \text{sign}(\mathbf{w}^T\mathbf{x}_2) = 1$, it must necessarily be the case that $\text{sign}(\mathbf{w}^T\mathbf{x}_3) = -1$. However $y_3 = 1$ and so Player 1 has a winning strategy.

Thus $\texttt{TourDim}(2) <= 2$. But it is trivial to verify that $\texttt{TourDim}(2) \geq 2$ as Player 2 always has a winning strategy with just 2 data points. Thus $\texttt{TourDim}(2) = 2$ and so from Theorem 2 it follows that $\mathcal{T}_2^n$ has a forbidden configuration of size 4. $\qquad\square$

**Proof of Lemma 4**

*Proof.* Consider the sub-tournament of four items $\{a_1, a_2, a_3, a_4\}$ labelled $\mathbf{T}$. If $\mathbf{T}$ corresponds to an acyclic graph, there is nothing to prove. Therefore only the case where $T$ contains at least 1 cycle is considered. Without loss of generality, let the items which form a cycle have indices $a_1, a_2$ and $a_3$. Let the corresponding embeddings of each item $i$ to be $[u_i, v_i]$. For $a_4$ to not violate the rank 2 assumption, it too must have a corresponding embedding by which its interactions with the existing items are defined. Let the embedding of $a_4$ be $[u_4, v_4]$. Without loss of generality, let $a_1 \succ_{\mathbf{P}} a_2 \succ_{\mathbf{P}} a_3 \succ_{\mathbf{P}} a_1$. Let the angle between the embeddings of items $a_1, a_2$ be $\alpha$ and items $a_2, a_3$ be $\beta$. Since $a_3 \succ_{\mathbf{P}} a_1$, we have $\alpha + \beta > \pi$. Now consider the angle between $a_1, a_4$ to be $\theta$. The following cases are possible.

$$\theta \in (0, \alpha) \implies a_1 \succ a_4 \succ a_2$$

$$\theta \in (\alpha, \alpha + \beta) \implies a_2 \succ a_4 \succ a_3$$

$$\theta \in (\alpha + \beta, 2\pi) \implies a_3 \succ a_4 \succ a_1$$

In none of these cases can the item $a_4$ succeed, or precede all of the 3 existing items. As the set of items chosen was arbitrary, the result follows. $\qquad\square$

**Proof of Theorem 5**
This theorems follows naturally from Equation 1 obtained as part of the proof for Theorem 6. Given Equation 1, it follows that $S_{i+k+1 \bmod(2k+1)} \succ S_i \succ S_{i+k \bmod(2k+1)}$.

**Proof of Theorem 6**

*Proof.* This proof in constructive in nature. Consider the set of items in the tournament to be represented as $a_i$. For every item $a_i$ in the tournament, define $S_i$ to be the maximal set of items associated with $a_i$ such that $|S_i| < n$ and $S_i$ satisfies the following

$$\forall a_k \in S_i, a_j \notin S_i, a_k \succ_{\mathbf{P}} a_j \text{ iff } a_i \succ_{\mathbf{P}} a_j$$

By definition $S_i$ is non-empty as it contains $a_i$ and is not equal to $[n]$ as $|S_i| < n$. It can be easily observed that the item to set mapping is an equivalence relation. Let the set of all unique constructed sets be named $Q = \{S_1, S_2, ...S_\ell\}$ for some $\ell$. Note that these sets are disjoint due to the equivalence relation present between the items and the sets in $Q$. Let $V_i$ be a vector corresponding to any one of the items in the set chosen arbitrarily. This vector representation cannot capture the exact probabilities between the items but has the following property.

$$\sin\theta > 0 \iff S_i \succ_{\mathbf{P}} S_j \text{ where } \theta \text{ is the counterclockwise angle between } V_i \text{ and } V_j$$

Consider an ordering of $Q$ determined by the corresponding $V_i$, such that the $V_i$ are in counterclockwise order. Consider the set of sets corresponding to $V_i$ that precede $V_1$ to be $Q_1$ and the set of sets which correspond to vectors preceded by $V_1$ to be $Q_2$. It is clear from Theorem 12 that the sets in $Q_1$ have a transitive relationship amongst them. The same applies for $Q_2$. Consider two consecutive sets (consecutive defined w.r.t to the angle between the representative vectors) $\{V_1\} \cup Q_1$, say $X_1$ and $X_2$. From the maximal property of the sets, we know that there must exist another set $X$ such that

$$X_1 \succ X \succ X_2$$

or

$$X_2 \succ X \succ X_1$$

We now introduce the term 'set separator' to refer to the set $X$ i.e, $X$ is the separator of $X_1$ and $X_2$. There is a transitive relationship among the items of $V_1 \cup Q_1$, therefore $X$ must lie in $Q_2$. Similarly, the separator of any two consecutive sets in $\{V_1\} \cup Q_2$ must lie in $Q_1$. Also, the set separator for a unique pair of successive sets must be unique. To prove this by contradiction, consider 2 pairs of successive sets in $Q_1$, $X_1, X_2$ and $X_3, X_4$. Let $Y \in Q_2$ be the separator for $X_1, X_2$ and $X_3, X_4$ such that $X_1 \succ X_2 \succ X_3 \succ X_4$. This leads to one of the forbidden configurations shown in Figure 4 being obtained for the subset $\{X_1, X_2, Y, X_3\}$. Since each pair of successive sets in $Q_1$ has a corresponding set in $Q_2$, and vice versa, the number of sets in $Q_1$ and $Q_2$ must be equal. Let $k$ be the number of items in $Q_1$. The total number of sets present in $Q$ must be $2k + 1$. Let the sets in $Q$ now be ordered as $\{S_1, S_2 \ldots S_{2k+1}\}$ such that the sets are in counterclockwise order w.r.t. their corresponding vectors. In the above construction, the set $Q$ is split using the vector $V_1$. Similarly, the set $Q$ can by split by using any vector $V_i$(Corresponding to set $S_i$). Since the proof above generalizes to any split, the following relationship between the sets holds true.

$$S_i \succ S_j \iff i - j \pmod{2k+1} \le k \tag{1}$$

**Proof of Part(a)** Using the claims proven in **??**, it is sufficient to prove that all the items belonging to a single set lie in the same half-plane. This is trivially true since all items in a set $S_i$ lie in the same half-plane split by when the plane is split by any of the items due to the maximal property of the sets.

**Proof of Parts (b) (c) (d))** We now construct the sets $A_1, A_2, \ldots, A_k, B_1, B_2, \ldots, B_k, C$. Let $C$ by the set $S_1$. Consider

$$A_i = S_{i+1}$$

and

$$B_i = S_{i+k+1}$$

Since $A_1, A_2, \ldots, A_k$ and $B_1, B_2, \ldots, B_k$ are consecutive sets which lie in the same half place when $S_1$ is used to split the plane, the transitive properties on the sets represented by $A_i$ and $B_i$ holds. From our construction of sets $A_1, A_2, \ldots, A_k$ and $B_1, B_2, \ldots, B_k$ and the properties of the sets $S_i$ proven above, parts (c) and (d) hold true. $\square$

**Proof of Theorem 7**

*Proof.* We show this using construction. A $\mathbf{T} \in \mathcal{T}_2^n$ with a cycle of any length, $k$, can be constructed as follows. Let the embedding corresponding to item $i_k$ be $[\cos\frac{2j\pi}{k}, \sin\frac{2j\pi}{k}]$. It is clear that $i_t^T A^{\mathrm{rot}} i_{t+1} > 0 \forall t \in 1..k - 1$ and $i_k^T A^{\mathrm{rot}} i_1$.

To show this model can represent a size 3 cycle with any fixed pairwise probabilities, fix an arbitrary triplet $\{i, j, k\}$. Let $q_\ell = \ln(p_\ell/(1 - p_\ell)) \; \forall \ell \in \{1, 2, 3\}$ and the embedding, $[u_i, v_i]$, corresponding to each item $i$. By setting $u_i = 1, v_i = 0, u_j = 0, v_j = q_1, u_k = -q_2/q_1$ and $v_k = -q_3$, the arbitrary triplet can be satisfied for all values of $p_\ell \neq 0.5, \ell \in \{1, 2, 3\}$. By definition of $\mathcal{P}$, values of $0.5$ are disallowed and the result follows. $\qquad\square$

**Proof of Theorem 8**

*Proof.* First we aim to prove that the optimal ranking for $\mathbf{T} \in \mathcal{T}_2^n$ is one of the $n$ cyclic shifts of one of the permutations of the items. Furthermore, the permutation of the items considered is a counterclockwise ordering the corresponding vectors, and is proven using induction. Using Theorem 6, we can partition the set of all items into $2k + 1$ groups. We can first show that the items of a single group must appear in consecutive positions in one of the optimal rankings. This is proven as follows.

Consider there exists an optimal ranking with items which belong to the same group not occurring consecutively. Consider two items belonging to the same group, which have items from other groups present in between them in the ranking. Consider these items to be $a_1, a_2$, with $a_1$ present above in the rankings. Consider the number of upsets that the two items are involved in to be $u_1$ and $u_2$. If $u_1 \leq u_2$, $a_2$ can be placed right after $a_1$ in the ranking, creating a better or equivalent ranking in terms of upsets. Similarly if $u_1 \geq u_2$, $a_1$ can be placed directly above $a_2$ in the rankings to create an equivalent or better ranking. Therefore there exists an optimal ranking which has all items in the same group consecutively.

This theorem is then reduced to finding a ranking of groups, which is proven using induction on $k$.

**Base Case**
Consider the base case with $k = 1$. Let there be 3 groups, $C, A_1, B_1$. We can say that the optimal ranking cannot be any of the following
$$CB_1A_1$$
$$A_1CB_1$$
$$B_1A_1C$$
since all three rankings can be made better by swapping the second and third ranked groups. Therefore the 3 possible optimal rankings are
$$CA_1B_1$$
$$A_1B_1C$$
$$B_1CA_1$$
which are cyclic shifts of each other.

**Inductive Step**
One property of rankings which is useful for the inductive step proof is as follows. Let there be $2k + 1$ groups $G = \{g_1, g_2 \ldots g_{2k+1}\}$. Label the optimal ranking with the condition that $g_i$ be placed first in the ranking as $R_i$. The ranking $R_i$ with $g_i$ removed must be the optimal ranking for $G \setminus \{g_i\}$. This can be shown using contradiction i.e, if there was a better ranking for $G \setminus g_i$, that ranking with $g_i$ appended to the front would be better than $R_i$.

We now assume the theorem is true for size $2k - 1$ instances and aim to prove for the same for size $2k + 1$ instances. Consider $g_1$ as the first group in the ranking. This creates a certain number of upsets, for the purposes of ranking the remaining groups, 2 of the remaining groups can be merged into a single group. This follows from the observation in the proof of Theorem 6 that each group also 'separates' two groups. This can be considered an instance of the size $2k - 1$ problem. Therefore the set of optimal rankings with $g_1$ as the first group in the rankings is made up of $g_1$ as the first group and a cyclic sweep of the remaining items to fill the remaining positions. Therefore the optimal permutation must be among the sets created by considering each of the $2k + 1$ groups as the first group in the rankings. Let $R_{i,j}$ represent the ranking which has group $g_i$ as the first group and the remaining groups present as a cyclic sweep from $g_j$. Consider the case of $R_{1,k}$. Let $x_i$ represent the number of items in group $g_i$. If $R_{1,k}$ is a better ranking than $R_{k,k+1}$, it implies that

$$\sum_{i=k}^{n+1} x_i > \sum_{i=n+2}^{2n+1} x_i \tag{2}$$

by considering the shift of $g_1$ in the two rankings. The difference in the number of upsets between $R_{1,2}$ and $R_{1,k}$ is given by

$$x_2(-x_k - x_{k+1} \ldots - x_{n+2} + x_{n+3} \ldots x_{2n+1}) + x_3(-x_k - x_{k+1} \ldots - x_{n+3} + x_{n+4} \ldots x_{2n+1}) \ldots$$

Using Equation 2, it can be seen that each of the above terms are negative for any $j \leq n + 1$, making $R_{1,2}$ the better ranking. Any $j > n + 1$ cannot be considered as an optimal ranking since the first group as per the ranking must precede the second(otherwise switching them would decrease the upsets). Since either $R_{1,2}$ or $R_{k,k+1}$(both counterclockwise orderings) is better than $R_{1,k}$ whenever $k \leq n + 1$($R_{1,k}$ cannot be the optimal ranking when $k > n + 1$), and since this can be generalised for any $R_{i,j}$, it is shown that one of the counterclockwise orderings of the items is the optimal ranking. This shows that a geometric sweep of items when embeddings are known provides an optimal ranking. Using Theorem 12, we it can be proven that R2R can find an optimal ranking without using the embeddings. Considering a cycle $a, b, c$ is found, let $A, B, C$ be defined as in Algorithm 2. From this it is clear that the counterclockwise ordering of arms will follow $A \succ B \succ C \succ A$, but the ordering within each set remains unknown. Since Theorem 12 can be applied on $A, B, C$ by setting $d$ as $c, a, b$ respectively, all 3 sets must be total orders (acyclic). Therefore, the counterclockwise sweep order is obtainable by performing a topological sort. Concatenating the topological sorts of $A, B, C$ should result in a permutation which corresponds to a counterclockwise sweep being created. This is done in Algorithm 2 since the recursive calls of the algorithm on sets $A, B, C$ will result in topological sorts being returned. Since the algorithm is able to find a counterclockwise sweep, and considers all cyclic shifts of the permutation, it is optimal. □

**Proof of Polynomial Time Complexity** The proof given here is stronger than the stated theorem, here we show that R2R has a polynomial running time for any instance. R2R first aims to find a cyclic triplet. This can be done in polynomial time in multiple ways. If a cyclic triplet is not found, the algorithm is polynomial time since a Topological sort is sufficient to obtain the optimal ranking. If a cycle is found, the recurrence relation, $T(n) = 3T(\frac{n}{3}) + P(n)$ where $P(n)$ is a polynomial which represents the time taken to check and compare the upsets of all $n$ cyclic permutations. By solving this recursive equation, it can be shown that $T(n)$ is polynomial in nature.

**Proof of Theorem 9**

*Proof.* This algorithm is based on Theorem 8. BR2R first tries to find a cyclic triplet, these 3 items must belong to the same block, $B$, since there cannot be cycles or items which belong to different blocks.

If a cycle is not found, the time complexity analysis and optimality is trivial. The optimality is guaranteed by the properties of Topological sorting, and the time complexity is polynomial in $n$.

If a cycle is found, the remaining items can be put into 3 groups. The first group is the set of items that precede all 3 items in the cyclic triplet. These items cannot belong to $B$ since one of the forbidden structures, as seen in 4, would be formed. This is also true for the second group, which consists of all the items which are preceded by all 3 items in the cyclic triplet. The third group of items consists of items which belong to the block $B$. Since the blocks are ordered, an optimal ranking can be constructed once the optimal rankings are known for each of the 3 groups by simply appending the rankings. Therefore we can construct the recurrence relationship for the time complexity of BR2R

$$T(n) = P(|B|) + T(n - |B| - |F|) + T(|F|)$$

where $|F|$ is the size of the group of items which precede all 3 items of the cyclic triplet. By assuming the $T(n)$ is not sub-linear

$$T(n) = P(|B|) + T(n - |B|)$$

Since $|B| >= 3$, $T(n)$ a polynomial, making BR2R a polynomial time algorithm. Since the ranking obtained by the above algorithm follows the block structure i.e, arms belonging to higher blocks are ranked higher, and optimal ranking is used to rank arms in the same block, the ranking over all arms produced must be optimal as well. □

**Proof of Lemma 10**

*Proof.* Consider the definition of $\mathbf{T} \in \texttt{Block-}\mathcal{T}_2^n$.

$$\left\{ \mathbf{T} \in \texttt{Block-}\mathcal{T}_2^n : \exists \mathbf{s}, \mathbf{u} \in \mathbb{R}^n, \text{ a partition } \{H_k\}_{k=1}^b \in [n] \text{ s.t} \right.$$

$$\left. \phi(T_{ij}) = \left\{ \begin{array}{l} u_i v_j - v_i u_j \text{ if } \exists \ell \text{ s.t } i, j \in H_\ell \\ s_i - s_j \text{ otherwise} \end{array} \right\} \right\}$$

The $\mathbf{T}$ defined above can be expressed in the form $Q - Q^T$ where $Q$ is

$$Q_{ij} = \left\{ \begin{array}{l} u_i v_j \text{ if } \exists \ell \text{ s.t } i, j \in H_\ell \\ s_i \text{ otherwise} \end{array} \right\}$$

The rank of $Q$ can be established as $2b$ as follows. Consider a sub-matrix of $Q$ defined as

$$Q_k = [Q_{ij}] \forall i \in [n] \forall j \in H_k$$

Each column of this matrix can be expressed as a linear combination of two vectors $U_k, S_k$ which are defined as

$$U_k = \left\{ \begin{array}{l} u_i \text{ if } i \in H_k \\ 0 \text{ otherwise} \end{array} \right.$$

$$S_k = \left\{ \begin{array}{l} s_i \text{ if } i \notin H_k \\ 0 \text{ otherwise} \end{array} \right.$$

A column whose index in the original matrix $Q$ is $j$ can be expressed as $U_k v_j + s_k$. Since each sub-matrix is of rank 2, $Q$ must be of rank $2b$. Since the rank of the sum of two matrices is upper bounded by the sum of the ranks of the two matrices, the rank of $\mathbf{T}$ must be less than or equal to $4b$. Therefore $\mathbf{T} \in \mathcal{T}_r^n$ for some $r \leq 4b$. $\square$

**Proof of Theorem 11**

*Proof.* To prove this theorem, we assume the matrix completion subroutine used in Algorithm 4 is the OPTSPACE algorithm of (16). From Lemma 10 we know that the rank of a local global model $\mathbf{P}$ with $b$ blocks is at most $4.b$ and this is also passed as an input to the Matrix completion routine. The proof of the theorem closely follows the proof of Theorem 13 in (22)). Specifically, from Equation 8 in the proof of the theorem, under similar sample complexity $O(nb \log(n))$ samples each compared $O(b \log(n))$ times and for any $0 < \epsilon < \frac{1}{2}$, the Frobenius norm difference of the completed matrix and the true low rank matrix can be bounded as follows:

$$\|\hat{\mathbf{M}} - \mathbf{M}\|_F^2 \leq \frac{n^2 \epsilon \min(\theta^2, \Delta^2)}{4} \leq \frac{n^2 \epsilon \Delta^2}{4}$$

where $\mathbf{M} = \phi(\mathbf{P})$

Under the condition that $\epsilon \leq \frac{\theta^2}{n^2 \Delta^2}$, this further can be bounded by

$$\|\hat{\mathbf{M}} - \mathbf{M}\|_F^2 \leq \frac{\theta^2}{4}$$

This implies that for every $(i, j)$ pair where $i$ and $j$ are in different blocks, it must necessarily be the case that $|\hat{M}_{ij} - M_{ij}| \leq \frac{\theta}{2}$. This implies then that in the reconstructed tournament using $\hat{\mathbf{M}}$,

$i \succ_{\mathbf{T}(\hat{\mathbf{M}})} j$ if and only if $i \succ_{\mathbf{T_P}} j$. This guarantees that the tournament $\mathbf{T}(\hat{\mathbf{M}})$ would be the same as $\mathbf{T_P}$ upto the blocks. Once the tournament is reconstructed, the algorithm runs the BR2B routine on $\mathbf{T}(\hat{\mathbf{M}})$. The BR2B algorithm starts by looking for a cyclic triplet if one exists. As a cycle cannot exist across blocks, the BR2R algorithm will necessarily find a cycle (if one exists) within a block. Thus the sets $S^+$ and $S^-$ would contain complete blocks in every recursive call. This further implies that the final concatenated ordering produced by the BR2R algorithm respects the blocks. Thus, the result follows. $\qquad\square$