# OpenReview forum: "On The Structure of Parametric Tournaments with Application to Ranking from Pairwise Comparisons"
_NeurIPS.cc/2021/Conference — NeurIPS 2021 Poster_

### Official Review · Reviewer_qX9G · 2021-06-28

**Rating:** 6
**Confidence:** 4

**Summary:**

The paper studies the minimum feedback arc set problem in certain classes of tournament graphs. In these classes, the edges of the tournament are defined by preference matrices of rank r. This study is motivated by arguing that such tournaments arise out of real-world data sets in various application domains.

The paper gives bounds on the size of forbidden configurations for rank-r tournaments. For rank 2 tournaments, the paper identifies the exact forbidden structures. The paper also studies a restricted class, called block rank 2 tournaments, which is claimed to subsume several practical models. It gives a polynomial-time algorithm for the minimum feedback arc set problem on instances of this class. Finally, it shows an application of their work to the problem of learning how to rank from pairwise comparisons and provide some experimental results.



**Limitations And Societal Impact:**

Yes.

**Main Review:**

The feedback arc set problem is certainly very important. However, I am not very convinced that the restriction considered (i.e., defined by such matrices) is very interesting in terms of applications. For instance, the results are applied to the so-called local-global preference model, which is defined in the paper, but whose importance is not really clear.

The writing is overall OK, but sloppy at times (see below).

Overall, the paper is good, but the motivation and importance of this specific problem/model are unclear.

Specific Comments:

Line 55: “See” should not be capitalized.
Line 64: “as” should be removed.
Line 74: “for the same” -- > “for the problem”.
Line 115: explain what “cdf” is.
Step 2 in Algorithm 1: a comma is missing before “(x_k, y_k)”.
Line 144: remove “as”. “w.r.t” -- > “w.r.t.”.
Line 152: “Inability” -- > “The inability”.
Line 156: a period is missing at the end of the line.
Line 181: “present the same”, please rephrase.
Line 183: What is “cone (U \cup V)”?
Line 219: a period is missing at the end of the line.
Line 257: extra space after “(“.
Lines 284 and 287: a period is missing at the end of each line.



**Time Spent Reviewing:**

12

---

> ### Author Response · Authors · 2021-08-06
> **Addressing reviewer comment on applicability and motivation**
>
> We would like to thank the reviewer for the careful reading of the paper and the specific comments on writing. We will address all these in the final version of the paper. We address the comment on the usefulness of the proposed model below:
>
> $Applicability ~and ~Motivation$
>
> Figures $1$ and $2$ show the motivations behind the proposed local-global model. Figure $1$ shows a representative set of real-world tournaments arising in various application domains that have a block structure that can be naturally modelled by the local-global model. The green shaded region in Figure $2$ visually represents the fraction of additional real-world tournaments (out of $290$ real-world tournaments from www.preflib.org) that the local-global model can represent which popular models like BTL/Thurstone cannot represent. Note that the local-global model subsumes the BTL/Thurstone models and so can represent the transitive tournaments in the blue region of Figure $2$ as well.
>
> To summarize, we believe the local-glocal model is important because of the following reasons:
>
>  (a) It is a  parametric model with just $3n$ parameters where $n$ is the number of items/nodes and so can be effectively estimated from pairwise data.
>
>  (b) It can model flexible intransitive tournaments in addition to transitive tournaments.
>
>  (c) Several intransitive real-world tournaments have block structures as mentioned above and hence can be modelled naturally using the local-global model.

---

> > ### Comment · Reviewer_qX9G · 2021-08-31
> > **Acknowledgment**
> >
> > I thank the authors for their replies. My score remains unchanged.

---

### Official Review · Reviewer_AqtE · 2021-07-16

**Rating:** 6
**Confidence:** 3

**Summary:**

This paper focus on minimum feedback arc set problem in tournaments (MFAST) and characterizes the relation between the preference matrix rank with forbidden configuration. The authors propose a novel polynomial-time algorithm on MFAST with preference matrix rank 2. The authors further extend the setting to a block rank 2 tournament, where nodes can be clustered into several blocks and in each block, the preference matrix has rank 2. The proposed algorithm can also work on the non-tournament cases with a matrix-completion step. The experiments shows that the proposed algorithm reaches better performance than other works, in both synthetic (non-tournament) and real-world datasets.

**Limitations And Societal Impact:**

 The authors adequately addressed the limitations and potential negative societal impact of their work

**Main Review:**

This paper tackles an important problem and the solution provides a novel view of the classical problem. Eventhough the most theoretical contribution is restricted by the preference matrix with rank 2, the experiemnts in a flexible setting show promising results.

Here are some questions and suggestions.
In the contribution, the authors claim that the proposed algorithm can solve MFAST problem with rank 2. However, in the synthetic experiments, the algorithm does not recover the optimal solution. Is this because the synthetic dataset is not a tournament? It might be good to include synthetic tournament experiments and analyzes the effect of preference matrix rank other than 2.

The performance metric is not normalizes, which makes it harder to compare across different datasets.

The algorithm 4 is named PairwiseBlockRank but Theorem 11 uses ParametricBlockRank.




**Time Spent Reviewing:**

5

---

> ### Author Response · Authors · 2021-08-06
> **Addressing reviewer comment on theoretical contributions and synthetic experiments**
>
> Thanks for the detailed review and the suggestions. We address the concerns raised in the review below.
>
>
> $Reviewer ~ comment ~ about ~ theoretical ~ contributions ~ restricted ~ to  ~rank ~ 2$:
>
> We note that our contributions extend beyond the rank-$2$ case. In Section $6$, we introduce the local-global model, where we partition the preference matrix into several rank $2$ blocks. This model is motivated by the fact that real-world tournaments often have a block structure (See Figures $1$ and $2$). In Lemma 10 we state that the rank of a block-rank $2$ matrix with $b$ blocks is $\leq 4b$. Moreover, as explained further below, our experiments are also not just on the rank $2$ case.
>
> $Synthetic ~experiments:$
>
> As the reviewer rightly points out, part of the reason why the proposed algorithm does not achieve optimal performance here is due to the fact that the synthetic data is not a tournament. This is indeed by the design of our experiment, where only a subset of pairs in the tournament are chosen and then sampled only a fixed number of times according to the underlying Bernoulli preference probability of the pair. Note that the sampling might flip the edge direction of a few compared pairs in the observed tournament. Thus the observed comparison graph is not only incomplete (and hence not a tournament) but also noisy. This experiment was done to demonstrate the robustness of the proposed algorithm in practical learning to rank scenarios where not all pairs can be compared and those that are compared are done so only for a limited number of times. Please refer to the usage of the parameters $K$ and known fraction($\beta$) as described in the $Synthetic ~Experiments$ subpart of section $8$.
>
> $Experiments ~ on ~ matrices ~ other ~ than ~ rank ~ 2:$
>
> As mentioned in lines $299$-$301$, the synthetic data experiments were conducted on a Block-rank $2$ matrix with $b=3$, which implies an upper bound of  $12$ on the rank of the skew-symmetric matrix. We wish to point out that the matrix completion routine was run not just with input rank $2$ but with ranks of $2$, $4$, $6$ and $8$ and the best rank was cross-validated using a $50:20:30$ train-validation-test split. We will make this clear in the experiments section.
>
> $Normalization ~of ~performance ~measure:$
>
> We have provided the exact number of upsets with respect to the test set in the paper. As suggested by the reviewer, we computed the normalized performance measure where the normalization is done w.r.t the number of unique pairs that appear in the test split.  The results are reported in the table below.
>
>
>  | Algorithm/Dataset     | MC + Copeland | MC + Borda | Rank Centrality | Blade-Chest | PairwiseBlockRank
> | ----------- | ----------- |----------- |----------- |----------- |-----------
> | Dota      | 0.311 (0.01)       | 0.368(0.03)  | $\mathbf{0.215(0.01)}$  | 0.228(0.01)  | 0.23(0.03)
> | Tennis      | 0.314 (0.005)       | 0.314(0.006)  | 0.285(0.008)  | 0.311(0.006)  | $\mathbf{0.281(0.006)}$
> | SushiA      | 0.036 (0.02)        | 0.18 (0.02)  | 0.042 (0.01)  | 0.06 (0.01)  | $\mathbf{0.034 (0.02)}$
> | SushiB      | $\mathbf{0.176(0.006)}$        | 0.188 (0.005)  | $\mathbf{0.176(0.006)}$ | 0.203 (0.006)  | $\mathbf{0.176(0.006)}$
> | Jester      | $\mathbf{0.05 (0.001)}$       | 0.23 (0.002)  | $\mathbf{0.05 (0.001)}$   | 0.15 (0.022)  |$\mathbf{0.05 (0.001)}$
>
>
> $Typo ~in ~Theorem ~11:$
>
> Thanks for catching this. It should indeed read as $PairwiseBlockRank$ in the statement of Theorem $11$.

---

### Official Review · Reviewer_fvcq · 2021-07-18

**Rating:** 7
**Confidence:** 3

**Summary:**

The paper studies real-valued input matrices to the minimum feedback arc set problem on tournaments (MFAST). The matrices are assumed to have low rank and originate from pairwise preference matrices under certain link functions. The authors present a geometrical condition that enables a polynomial-time algorithm to solve MFAST for rank-2 matrices. They apply their algorithm in experiments on learning to rank from pairwise comparisons.

**Limitations And Societal Impact:**

The paragraph on broader impact is adequate.

**Main Review:**

Originality/Significance:
The results appear to be original and are very interesting from a complexity perspective. The proposed model and algorithm contribute significantly to both the optimization of the MFAST and tractable modeling of ranking from pairwise comparisons in practice.

Quality:
The writing is of high quality. Due to very limited time I could not check the technical details and thus cannot judge correctness of the claims.

Clarity:
The paper is quite clear overall. Unfortunately, the authors did not number the references in the bibliography although they use number citations (they seemed to be in the right order, however).

Questions:
It would be great if the authors could point out how their results depend on the link function. In other words, are the claims still true if we simply assume an arbitrary skew-symmetric rank-2 matrix $M \in \mathbb R^{n \times n}$ ? In this case the results would be even more interesting.

**Time Spent Reviewing:**

3

---

> ### Author Response · Authors · 2021-08-06
> **Addressing reviewer question on the generality of our results**
>
> Thanks for the review and the positive comments about our work. We address the main question raised by the reviewer below.
>
> $Question ~ about ~ our  ~ results  ~holding ~ for ~ arbitrary ~ skew ~ symmetric ~ matrices $:
>
> Our results indeed hold for any skew-symmetric rank-2 matrix as long as it does not contain zero in the non-diagonal entries. This condition is necessary to associate a skew-symmetric matrix to a tournament. Given such a skew-symmetric matrix, it can be converted to a pairwise preference matrix by applying the inverse link function. Note that the specific link function chosen would show up in the sample complexity results of learning to rank using pairwise comparisons.  However, our results on the structural/geometric characterization of rank $2$ tournaments are completely independent of the link function chosen. We have also mentioned this in lines 119-120.
>
> $Comment ~ about ~ references$:
>
> Thanks for catching this. We will fix the bibliography to include the number in the references.

---

### Decision · Program_Chairs · 2021-09-27

**Decision:**

Accept (Poster)

**Comment:**

The reviewers are satisfied with the author responses and agreed with acceptance. The authors incorporate reviewer feedback and additional experiments presented in the rebuttal in the final manuscript.